# Diseasomics: Actionable machine interpretable disease knowledge at the point-of-care

**Asoke K. Talukder**[1,2], **Lynn Schriml**[3], **Arnab Ghosh**[4], **Rakesh Biswas**[5],
**Prantar Chakrabarti**[6,7], **Roland E. Haas**[8] *

**1** SRIT India, Bangalore, India, **2** Computer Science & Engineering, National Institute of Technology Karnataka (NITK), Surathkal, India, **3** University of Maryland School of Medicine, Maryland, United States of America, **4** Indian Institute of Technology Bombay, Mumbai, India, **5** Kamineni Institute of Medical Sciences, Narketpalle, Telangana, India, **6** Vivekananda Institute of Medical Sciences, Kolkata, India, **7** Cybernetic Care, Bangalore, India, **8** International Institute of Information Technology Bangalore (IIIT-B), Bangalore, India

* roland.haas@iiitb.ac.in

**Data Availability Statement:** Biomedical data used in this study can be found in S1 Text, all other relevant data is provided in the supplementary files.

## Abstract

Physicians establish diagnosis by assessing a patient's signs, symptoms, age, sex, laboratory test findings and the disease history. All this must be done in limited time and against the backdrop of an increasing overall workload. In the era of evidence-based medicine it is utmost important for a clinician to be abreast of the latest guidelines and treatment protocols which are changing rapidly. In resource limited settings, the updated knowledge often does not reach the point-of-care. This paper presents an artificial intelligence (AI)-based approach for integrating comprehensive disease knowledge, to support physicians and healthcare workers in arriving at accurate diagnoses at the point-of-care. We integrated different disease-related knowledge bodies to construct a comprehensive, machine interpretable diseasomics knowledge-graph that includes the Disease Ontology, disease symptoms, SNOMED CT, DisGeNET, and PharmGKB data. The resulting disease-symptom network comprises knowledge from the Symptom Ontology, electronic health records (EHR), human symptom disease network, Disease Ontology, Wikipedia, PubMed, textbooks, and symptomology knowledge sources with 84.56% accuracy. We also integrated spatial and temporal comorbidity knowledge obtained from EHR for two population data sets from Spain and Sweden respectively. The knowledge graph is stored in a graph database as a digital twin of the disease knowledge. We use node2vec (node embedding) as digital triplet for link prediction in disease-symptom networks to identify missing associations. This diseasomics knowledge graph is expected to democratize the medical knowledge and empower non-specialist health workers to make evidence based informed decisions and help achieve the goal of universal health coverage (UHC). The machine interpretable knowledge graphs presented in this paper are associations between various entities and do not imply causation. Our differential diagnostic tool focusses on signs and symptoms and does not include a complete assessment of patient's lifestyle and health history which would typically be necessary to rule out conditions and to arrive at a final diagnosis. The predicted diseases are ordered

**Funding:** The authors received no specific funding for this work.

**Competing interests:** The authors have declared that no competing interests exist.

according to the specific disease burden in South Asia. The knowledge graphs and the tools presented here can be used as a guide.

## Author summary

A doctor at the point-of-care is expected to have the complete medical knowledge with latest updates in evidence-based medicine (EBM). The doctor is also expected to use this complete knowledge accurately during a medical interaction with a patient. In reality this is not the case—there are gaps in knowledge acquisition and gaps in clinical decision making. To address these gaps, in the past AI based expert driven rule-based clinical decision support systems were developed. Rule-based systems are rigid and often fail in case of complex diseases. We therefore built an AI based evidence driven clinical decision support system. We mined PubMed, Wikipedia, textbooks, medical records, etc. to extract clinical knowledge. We used this clinical knowledge as glue to connect ontologies to construct a machine interpretable antireductionistic diseasomics knowledge graph. The diseasomics knowledge graph is stored in a Neo4j property graph database in a cloud for online and realtime access using JSON-RPC API and works like the physicians' brain digital twin. We used the digital triplet node2vec techniques to mine unknown knowledge and to create a learning healthcare system. The integrated diseasomics knowledge system is available for use at https://triage.cyberneticcare.com/diseasePrediction.

## Introduction

There is global consensus on the goals of universal health coverage. Universal health coverage has been defined as "all people receiving quality health services that meet their needs without exposing them to financial hardship in paying for them" [1]. A study covering 183 countries finds that strengthening the health workforce is the major hurdle towards achieving universal health care [2].

Strengthening of the health workforce will be possible by increasing the number of skilled health workers and/or reducing the knowledge-gap between specialist-care providers in secondary/tertiary urban hospitals and non-specialist rural community health workers at the primary care through the use of Artificial Intelligence (AI). A consensus study from the National Academy of Medicine [3] has revealed that medical record documentation and coding requirements are one of the main reasons of physician burnouts. AI can help to reduce this workload and mitigate the risk of burnouts [4,5].

In AI, we classify an object through its features often learned from the data. In medical diagnoses, signs, symptoms, findings of the current health problem, age, sex, lifestyle, and disease history of the patient (and family) can be considered as features of the illness without any loss of generality. There are many roadblocks for the use of AI on medical data such as the availability of digital data, the lack of interoperability of coding systems, acceptance, lack of transparency, unexplainable outcomes, bias, cybersecurity concerns, and disappointments because of unrealistic promises. But a lot has changed and the tremendous progress both in AI as well as in the digital transformation of healthcare has opened new possibilities and application domains [4,5]. An important factor is that the machine readability and interpretability of medical data has advanced from the earliest handwritten descriptions to fully machine and human readable and interpretable digital medical data [6,7].

Medical decision making is a complex process and can be viewed along a spectrum, with categorical (or deterministic) reasoning at one extreme and probabilistic (or evidential) reasoning at the other extreme [8]. Categorical medical decision-making processes follow single flowchart, whereas, probabilistic decision making involves multiple overlapping flowcharts with added uncertainties. Numerous diagnostic decision, clinical decision and differential diagnosis support systems address this complexity, aiming to provide effective tools to guide disease characterization, enhance patient safety through a reduction of medication errors, improve clinical guidelines adherence, and identify relevant imaging analysis based on the patient's symptoms matched to diagnoses [9,10,11]. Tools are developed both commercially (e.g., visualDx [10,12,13,14], AMBOSS [15], Ada Health [16]) and in clinical laboratories (e.g., DXplain [17,18]) to integrate medical images, explore potential etiologies in literature or characterize electronic medical textbooks for sets of human diseases. Artificial Intelligence (AI) information systems provide novel tools in the diagnostician's toolkit, towards more informed clinical decisions and broadening the availability of diagnosis support in rural communities [19,20].

To address the challenges of offering an interoperable, openly available differential diagnosis tool across the breadth of human diseases, we have developed the 'Diseasomics' tool that combines biomedical ontologies, knowledge from published data and EHR records to provide a disease knowledge, pharmacogenomic knowledge bases regularized and connected via biomedical ontologies and controlled vocabularies. The Symptom Ontology (SYMP), Human Disease Ontology (DO) [21], SNOMED CT (Systematized Nomenclature of Medicine–Clinical Terms) [22,23], UMLS CUI (Unified Medical Language System, Concept Unique Identifier) [24], ICD10 CM (International Classification of Diseases 10th Revision) [25], HGNC (Human Genome Organization Gene Nomenclature Committee) [26], genes, and pharmacogenomics (PharmGKB) [27] enable interoperability and foster reuse across diagnostic systems.

Many of these ontologies and knowledge networks are curated by experts representing 100s of person-years of knowledge derived from evidence or extracted from millions of patients' health data. The majority of this knowledge is available for download. Internet search engines such as 'Google' can be used to locate sites containing medical or clinical information from the original knowledge sources. However, choosing the right search keywords or choosing the relevant answer from a large result set requires domain knowledge [5], which is often missing in resource constraint point-of-care. Therefore, making this machine interpretable diseasomics knowledge available and accessible to health workers and beneficiaries at the point-of-care 24/7 through a simple user interface will immensely improve the accuracy and timeliness of the care, especially in resource limited setups or care for disadvantaged population groups.

One critical component in evidence-based and networked medicine [28] is the diagnosis of a disease from the patient's signs, symptoms, findings, and co-existing conditions. However, there is limited work in this area of machine interpretable knowledge. The 'Symptom Ontology' provides the ontology of many symptoms but lacks the cross-references to diseases [29]. Xia et al. mined literature to identify and retrieve symptom to disease relationships and build a symptom-disease knowledge graph [30]. They identified alignment of 8,514 disease concepts from the Human Disease Ontology and symptoms from the Symptom Ontology to 15,970,134 MEDLINE/PubMed citation records [30]. Because this knowledge is mined from biomedical literature, it is biased towards complex and severe diseases, or diseases having severe conditions. Zhou et al. constructed a symptom-based human disease network of 134 diseases and 316 symptoms through the diseases and their underlying molecular interactions [31]. This network is biased towards genetic disorders. Rotmensch et al. constructed a disease-symptom network of 156 diseases and 491 symptoms from emergency department patient visits, which is biased towards emergency care [32]. The above knowledge constructed by Zhou et al. [31] and

Rotmensch et al. [32] are available in the open domain, therefore we semantically integrated them with the current diseasomics knowledge graph. The 'Diseaseome tool' has improved upon previous disease-symptom relation mining work by: (1) utilizing the DO's expanded disease classification of 10,579 disease concepts (2021 version of the DO); (2) including more exquisite relationship with weights between entities for knowledge graph construction. This significant expansion of the number of diseases represents diseases across all disease types; (3) all systems [12,15,17,30,31,32] described above are designed for human consumption only, whereas the diseasomics presented here is both human understandable and machine interpretable; (4) because the diseasomics knowledge is machine interpretable, other ontology based knowledge sources or knowledge extracted using AI techniques can easily be integrated semantically or thematically. Supplementary Information S2 Text shows how diseasomics is integrated with spatial comorbidity, resistomics, patholomics, oncolomics, EHR, and a secured telemedicine system. (5) diseasomics is stored in a graph database and can be accessed through API (Application Programming Interface) for the use of a health workers at the point-of-care.

Some Internet search engines can perform English type rudimentary natural language questions and answers. Although a Google search can provide a list of sites to the questions like "symptoms of dengue", it is not able to answer questions like "disease associated with nausea and rash". Such search engines are not effective to do an acceptable differential diagnosis or patient stratification with multiple symptoms. Our diseasomics disease-symptom knowledge network can easily be used for symptom-disease query and differential diagnosis.

## Methods

### The diseasomics architecture

Fig 1 shows the architecture of the diseasomics knowledge graph. The knowledge graph is stored in a Neo4j properties graph database [33,34]. It includes data from both ontologies and

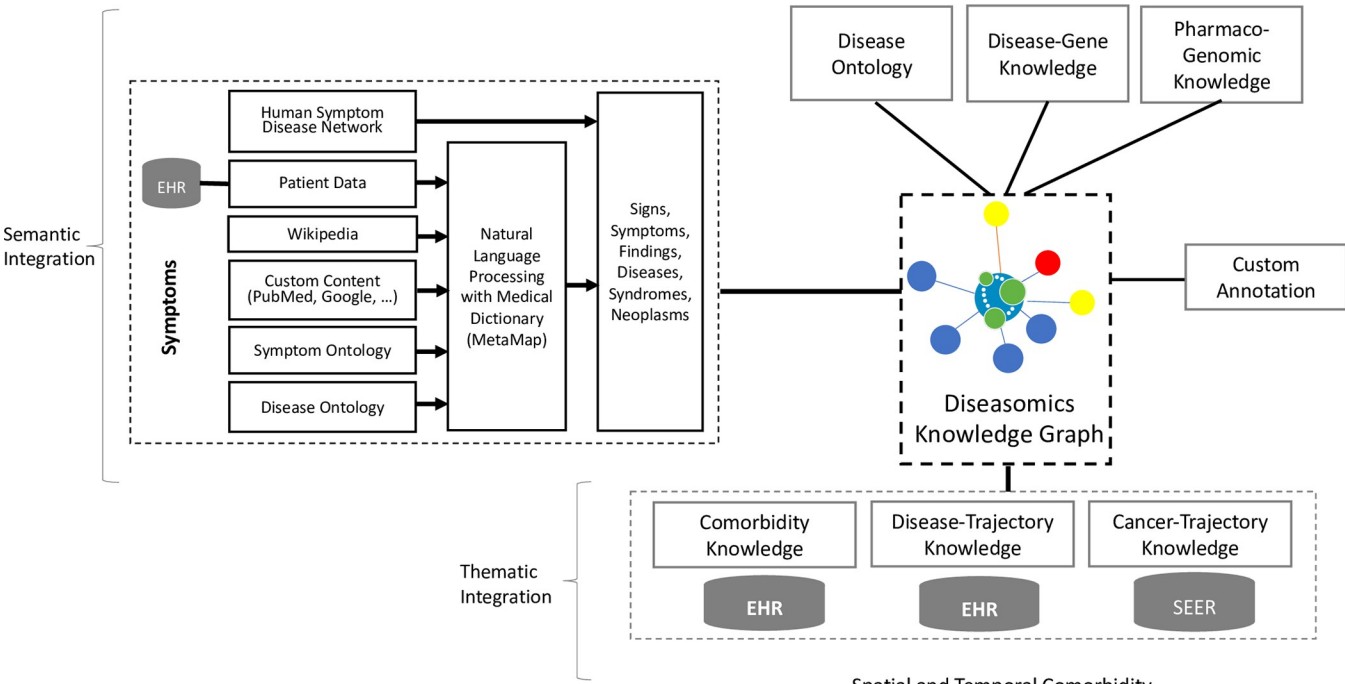

**Fig 1. The Diseasome Knowledge Graph Architecture.**

patient data from EHR (see S1 Text). We used ontology data from Symptom Ontology, Disease Ontology, SNOMED CT, ICD10 CM, and UMLS. We used data from the disease-gene network (DisGeNET) [35] and the pharmaco-genomics network (PharmGKB). We also used Wikipedia and PubMed knowledge sources. In addition, we used external knowledge sources like textbooks and manually curated content grouped under CustomContent. For tuning of the knowledge graph and customization purposes we used whitelists and blacklists to add or remove knowledge in the CustomAnnotation category.

The advantage of a graph database over a relational database is its maintenance and navigational flexibility. A user can enter the graph from anywhere and exit from anywhere. For example, the user can move from a symptom to a gene or just access the disease trajectory. Even the user can find the disease prevalence in an age group through spatial comorbidity. Moreover, knowledge can be added in a step-by-step fashion and made richer by the day. Furthermore, a graph database facilitates mining of hidden knowledge from the digital twin [36]. This flexibility makes the knowledge graph suitable for differential diagnosis, medical education, confirmatory diagnosis, referrals, research, or even cybersecurity.

## Symptom-disease network

We examined the symptom ontology (SYMP) data [29] that has 860 active terms with 369 ICD9CM_2005, 175 UMLS_CUI, and 368 UMLS_ICD9CM_2005_AUI terms. The Disease Ontology (DO) annotations in 'doid.obo' for certain diseases include limited disease symptoms. To construct a dependable symptom-disease relationship, we need accurate and complete symptoms for each disease which is missing in the SYMP or DO. Therefore, we included knowledge sources outside of 'symp.obo' and 'doid.obo' ontologies. We analyzed the knowledge available in PubMed [37] and Wikipedia [38]. Our research suggests that Wikipedia contains encyclopedia or textbook-type symptomatology. This is very useful in primary care settings; whereas, PubMed contains specialized focused research knowledge related to diseases, organ systems, prognosis, and therapeutics that are useful in secondary or tertiary care. We, therefore, looked at this critical area and added knowledge from Wikipedia. We, however, included PubMed and symptomatology as a secondary knowledge source through the custom content as shown in Fig 1. We combined additional knowledge of the molecular human symptom-disease network [31] and the symptom knowledge obtained from emergency room patient data [32] to build a comprehensive symptom-disease network covering infectious diseases, non-communicable diseases, molecular diseases, and acute care.

We used the Python Wikipedia-API [39] to access the Wikipedia knowledge base for each disease in the DO. We considered PubMed and symptomology textbook knowledge as secondary source of knowledge. For example, there is no Wikipedia page for 'benzylpenicillin allergy' (DOID:0040003). PMID:14483916 referred in DOID:0040003 contains description of penicillin allergy but not sufficient to extract symptoms of the allergy. Therefore, we added penicillin allergy from PMID:22247826 [40] with respect to 'DOID:0040003 benzylpenicillin allergy' as an addendum (CustomContent) to Wikipedia. This combined content of DO, SYMP, Wikipedia, PubMed, and CustomContent is processed using the natural language processing (NLP) engine MetaMap [41].

The human symptom disease network (HSDN) [31] contains 11,482 relationships (rows) with 316 symptoms and 134 diseases that used the MeSH terminologies [42] for both disease and symptoms. We converted the MeSH symptoms terms into UMLS CUI codes and the MeSH disease terms into DO disease terms. The structured EHR data of the emergency room disease-symptom data is published as a human understandable pdf file [32]. For this data, we

converted the disease into DO codes and symptoms into UMLS CUI codes. We integrated both these disease-symptom knowledge as shown in Fig 1.

This preparation work of converting, integrating, filtering, and curating helped us to construct a corpus of terms that mostly relate to infectious, non-communicable diseases, molecular diseases, and acute care. A total of 84,506 UMLS CUI terms were identified by MetaMap. Out of these terms, a total of 31,440 unique UMLS CUI codes were obtained. We included terms related to 'Sign or Symptom', 'Disease or Syndrome', 'Neoplastic' and 'Finding'. We performed lemmatization on these terms. The goal of stemming and lemmatization is to reduce inflectional forms and sometimes derivationally related forms of a word to a common base form [43]. WordNet [44] is a rich stemming and lemmatization database for English language. However, WordNet works on a single English word (unigram). Symptoms are generally a continuous sequence of multiple words (n-words or n-gram) forming one meaningful medical concept. Therefore, we implemented our own lemmatization algorithm over and above Meta-Map stemming and lemmatization algorithms. Our intention is to make the machine agnostic to the context and coding nomenclatures used by different ontologies. For example, 'weight loss' can map to one of the 11 following unique UMLS CUI terms:

C0041667:Excessive body weight loss (finding)
C0586746:Excessive weight loss (finding)
C0936227:Abnormal decrease in weight (finding)
C1262477:Progressive weight loss
C1285591:Weight loss (amount) (observable entity)
C1303041:Weight loss from baseline weight
C1542746:Abnormal weight–loss–symptom
C1828173:Unexplained weight loss (finding)
C1961006:Recent weight loss (finding)
C2363736:Unintentional weight loss (finding)
C5203233:Weight loss

All these 11 UMLS terms are lemmatized to the machine interpretable UMLS CUI concept 'C1262477' and human understandable term 'Weight loss'.

Following the removal of duplicate CUI codes, stemming, and lemmatization unique UMLS CUI entries were identified. All concepts that are present in both UMLS and SNOMED were included resulting in 11,188 UMLS concepts. These 11,188 concepts were mapped with the 10,597 diseases in the DO.

## Disease-disease network

We used the doid.obo disease ontology version 2.6.2 downloaded on 26 January 2021. The DO has totally 13,046 terms with 2449 obsolete terms. The DO file has 8,565 terms that include the annotation in the 'def' (definition) field. It has a total of 35,540 xref entries, that include 16 types of cross-references (see Table 1).

In the DO file, there are 3652 disease terms that contain ICD10CM cross references. There are 5043 disease terms with SNOMED CT cross-reference, and 6871 disease terms with UMLS CUI cross-reference codes. We included all ICD10CM, SNOMED CT and UMLS_CUI terminologies. For 10,597 disease nodes we got 3516 entries that contain one or more ICD10 codes. The number of unique ICD10 codes used in the DO is 1542. ICD10 codes used in a clinic or EHR are explicit, such as "H54.1131: Blindness right eye category 3, low vision left eye category 1", whereas in the DO or the disease trajectory–because it is the definition of a disease instead of an instance of the disease–the ICD codes are generic codes at a higher hierarchical level like "H54: Blindness and low vision" instead of "H54.1131".

**Table 1. Types of cross-references.**

| Acronym | Meaning |
|---|---|
| EFO | Experimental Factor Ontology |
| GARD | Genetic and Rare Diseases |
| ICD10CM | ICD10 Clinical Modification |
| ICD9CM | ICD9 Clinical Modification |
| ICDO | International Classification of Diseases for Oncology |
| KEGG | Kyoto Encyclopedia of Genes and Genomes |
| MEDDRA | Medical Dictionary for Regulatory Activities Terminology |
| MeSH | Medical Subject Heading |
| NCI | National Cancer Institute |
| OMIM | Online Mendelian Inheritance in Man |
| ORDO | Orphanet Rare Disease O*ntology* |
| SNOMEDCT_US_2018_03_01 | SNOMED Clinical Terminology US Edition 2018 March |
| SNOMEDCT_US_2019_09_01 | SNOMED Clinical Terminology US Edition 2019 September |
| SNOMEDCT_US_2020_03_01 | SNOMED Clinical Terminology US Edition 2020 March |
| SNOMEDCT_US_2020_09_01 | SNOMED Clinical Terminology US Edition 2020 September |
| UMLS_CUI | Unified Medical Language System Concept Unique Identifier |

### Disease-gene network (DisGeNET)

For the disease-gene network we used the curated 'curated_gene_disease_associations_tsv'. In this file, there are 9703 genes and 11,181 diseases with 84,037 relationships. The complete data were used to construct the disease-gene network. For the DO-DisGeNET interconnection, we extracted all genes related to the 4527 UMLS CUI codes present in DO as term properties and established links. Gene names in the DisGeNET are gene symbols. To make the analysis universal, we included the HGNC standardized gene names as well.

### SNOMED CT network

SNOMED CT is machine interpretable, comprehensive, scientifically validated multilingual clinical healthcare terminology knowledge. It enables consistent representation of clinical content in electronic health records for seamless exchange of health data. SNOMED CT knowledge was imported into the Neo4j graph database by Campbell et al. [45]. We have used the 2019 release of SNOMED and have installed the entire SNOMED CT data in the Neo4j graph database using the Neo4j SNOMED documentation. To connect the DO with SNOMED, we used 4658 DO terms that have xref links to 4897 SNOMED SCTID.

### ICD10 CM networks

We have 1542 ICD10 CM unique codes used in the DO. These codes were used to connect DO with spatial comorbidity and temporal comorbidity from Spain and Denmark EHR, respectively.

### PharmGKB

The PharmGKB (Pharmaco-Genomics Knowledge Base) helps users in understanding how variations in a person's genetic makeup affects the patient's response to a drug. We used the "drugLabels.tsv" PharmGKB file in TSV format. To connect the PharmGKB with the main graph we connected it to 766 chemicals (or drugs) through 142 common genes.

### Disease spatial comorbidity network

For the construction of the diseasomics knowledge graph, we so far (previous steps) have considered the semantic integration of ontology data and disease related generic knowledge from various sources that do not include disease factors like age or sex. The diseasomics was extended with population specific thematic knowledge or country specific knowledge from patient data recorded in EHR. We have integrated two types of comorbidities data for this exercise, namely spatial comorbidity knowledge and temporal comorbidity knowledge. Spatial comorbidity is defined as co-occurring diseases in a person at a point-in-time, whereas temporal comorbidity is defined as co-occurring diseases in a person over a period-of-time.

To construct the spatial comorbidity network we used raw EHR data for all admissions in public hospitals during 2016 in Madrid, Spain published by the NHS authorities and downloaded from [46]. Each row in the raw data corresponds to an admission and the final diagnosis as described in Talukder et al. [7]. The data was available in 3 columns, namely, 'sex' (1 for male, 2 for female). 'age', and the 'diag' (diagnoses). In diagnoses all diseases were in ICD10CM codes listed in one column. We divided the patients into 18 age groups, namely 0–9, 10–19, 20–29, 30–39, 40–49, 50–59, 60–69, 70–79, 80–120 for male and female populations. This downloaded data was converted from Spanish into English. We used statistical techniques to select only those co-occurring diseases with p-value < 0.05 [7].

### Disease temporal trajectory network

For temporal comorbidity we used processed data that includes the entire spectrum of diseases covering the whole population of Denmark conducted by Jensen et. al [47]. The study includes 6.2 million patients with 65 million medical encounters over 14.9 years. This temporal trajectory data is available in Microsoft xls format with 4014 rows and 10 columns. We added the entire trajectory knowledge. However, to connect with the DO, we took only the intersection of the ICD10 codes that are present in the DO and the trajectory data. There are some diseases in the DO that are at the parent level like DOID:9351 (diabetes mellitus) which has a group of ICD10CM codes 'E08-E13'. These have been properly expanded and linked to multiple ICD10 nodes before connecting to the disease trajectory.

### Node embedding

Node embedding transforms a high-dimensional graph data into a low-dimensional representation where the structural information of the original graph is preserved. Graph embedding helped us construct the digital triplet of the disease knowledge. For vector embedding of the knowledge graph we used 'node2vec' on the disease-symptom network. 'node2vec' is an unsupervised learning algorithm that performs node embeddings. These embeddings are learned from the graph in such a fashion that nodes that are semantically close in the original knowledge-graph remain close in the embedding space. These properties of embedding are used as features in building machine learning models for various downstream tasks like link prediction and node labelling [48]. We used random walk graph embedding node2vec [49] to validate the knowledge graph and discover unknown knowledge. We took the diseasomics disease-symptom graph and converted it into a networkX graph. We then used this graph for graph embedding using node2vec in Python [49].

## Results

The diseasomics knowledge graph is a multigraph with multiple types of nodes and relationships (75,642 nodes without SNODMED CT and 753,746 relationship edges without

**Table 2. Diseasomics Nodes Statistics.**

| Node Type | Node Count |
|---|---:|
| Disease Nodes (DO) | 10,597 |
| UMLS Disease Nodes (DisGeNET) | 14,187 |
| SNOMED CT (ObjectConcept) | 474,074 |
| Genes | 9,703 |
| UMLS Symptoms (Signs, Symptoms, Findings, Disease) | 11,305 |
| ICD10 | 1,574 |
| Disease Trajectory | 681 |
| Spatial Comorbidity | 11,822 |
| PharmGKB | 766 |

SNOMED CT). Table 2 lists the nodes count for various types of nodes in the network. Table 3 shows the edges or relationships in the knowledge graph.

Fig 2 shows the graph of 'disease of metabolism' (DOID:00145667) with pathlength 2. This graph includes DO, SNOMED (Object Concept), ICD10, UMLS, Symptom, and Trajectory.

## Differential diagnosis

Using the symptom-disease graph and set algebra we derived the differential diagnosis from the knowledge available in the diseasomics. We use multiple symptoms and determine the likely disease from a large set of possible diseases. For multiple symptoms, we identify the common co-occurring conditions that are behind various signs, symptoms or findings. The diseasomics knowledge graph is intended to be used by non-specialist health workers at a resource-limited point-of-care. For complex cases, the health worker is instructed to refer the case to a secondary or tertiary care facility or an ambulatory service.

Fig 3 shows the differential diagnosis for three symptoms namely "fever", "coughing blood", and "night sweat". The symptom "fever" maps to two UMLS CUI codes namely, C0015967 and C0424755. These two codes have been lemmatized to C0424755. UMLS CUI code C0424755 is associated with 474 diseases. Coughing blood UMLS CUI code C0019079 is associated with 36 diseases. Night sweat with UMLS CUI code C0028081 is associated with 44 diseases. When we perform a set intersection of these three disease sets, we get three diseases, namely, DOID:552 pneumonia, DOID:399 tuberculosis, and DOID:117 heart cancer (Fig 3). The web link available in the 'def' field of the DO is converted into a live link that a user can click to go to the web site for detail knowledge about the disease.

**Table 3. Diseasomics Edges Statistics.**

| Edge Type | Edge Count |
|---|---:|
| SNOMED CT | 6,125,802 |
| DO-ICD | 2,418 |
| DO-IS-A | 14,567 |
| DO-SNOMED | 5,041 |
| DO-UMLS (DisGeNET) | 5,045 |
| DO-UMLS (Symptoms) | 59,467 |
| DO-Comorbidity | 2,747 |
| Disease Trajectory | 11,582 |
| Comorbidity | 65,151 |
| PharmGKB | 1,111 |

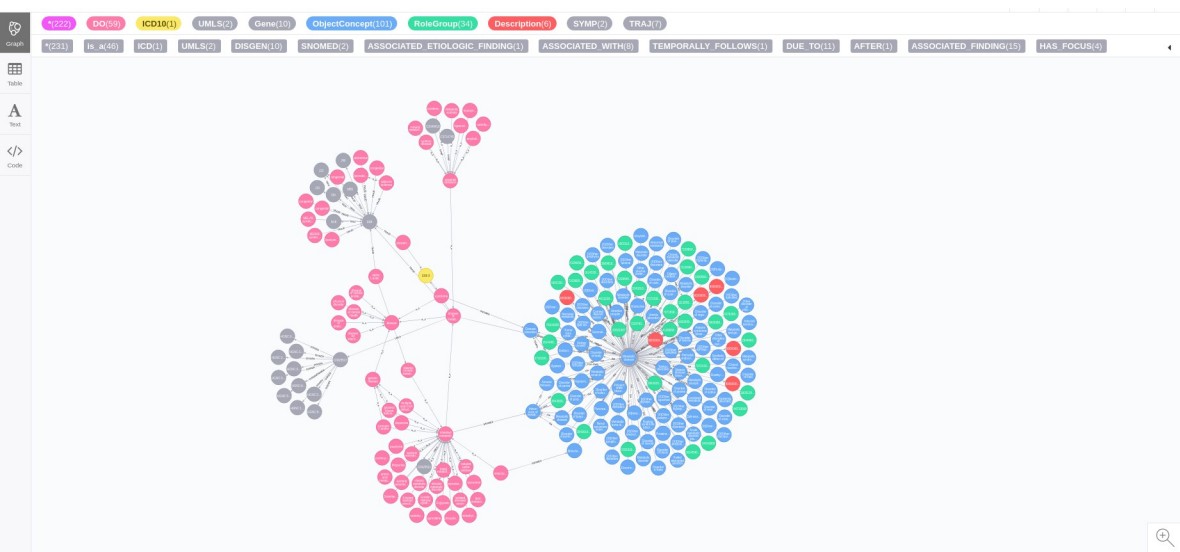

**Fig 2. The 'disease of metabolism' (DOID:0014667) with pathlength 2.**

## Validation of the knowledge graph

We performed the validation of the diseasomics knowledge graph in different ways. For validation of SNOMED CT, Trajectory, and DisGeNET we had selected about 50 nodes randomly and tested them manually one by one.

For differential diagnosis validation, we used symptoms as listed in the symptom ontology file symp.obo. We used MetaMap [41] to get the UMLS CUI codes for all symptoms in symp. obo. This gave us 65 different types of UMLS codes like [Amino Acid, Peptide, or Protein, Enzyme], [Biomedical or Dental Material], [Body Location or Region], [Body System], [Cell],

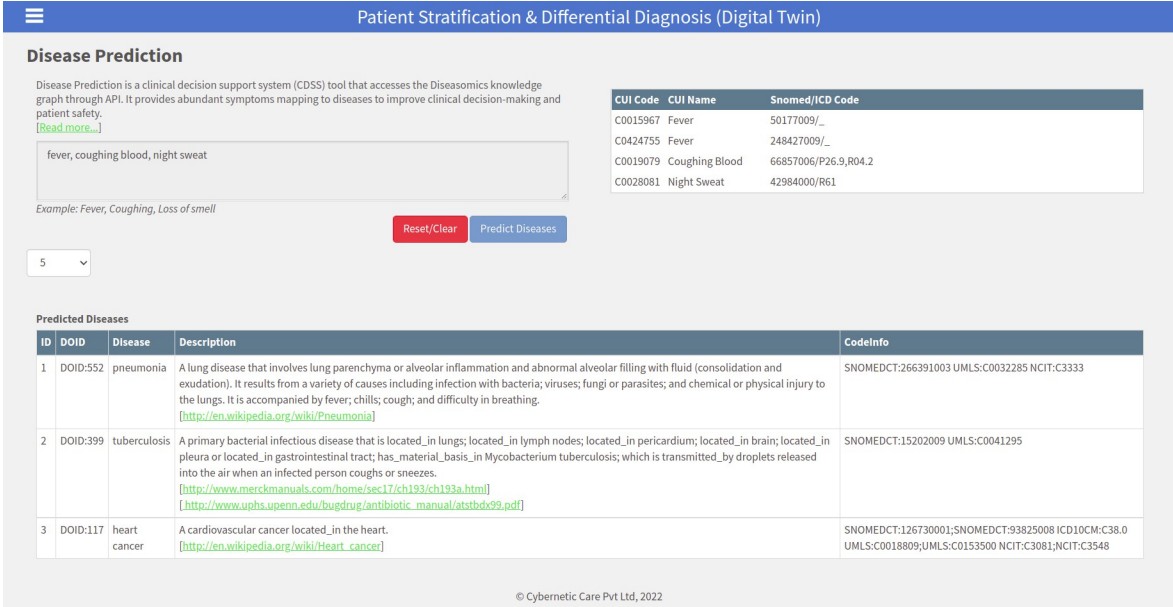

**Fig 3. The differential diagnosis for three co-occurring symptoms "fever", "coughing blood", "night sweat".**

**Table 4. Results of the differential diagnosis 2- and 3-symptom predictions.**

| Symptom combination is associated with a disease | DDx Correct | DDx Incorrect |
|---|---|---|
| Positive | 805 (TP) | 85 (FP) |
| Negative | 11 (TN) | 64 (FN) |

[Pathologic Function], [Sign or Symptom], [Tissue] etc. From these UMLS codes we selected only those UMLS CUIs codes that map to 'Sign or Symptom', which gave us 240 'Sign or Symptom' that were used for validation.

In our first approach, we performed 50 random draws of two symptoms and 50 random draws of three symptoms. These 100 combinations of symptoms (50 two symptoms + 50 three symptoms) were given to three certified clinicians for blind validation in a city near National Institute of Technology Karnataka, Surathkal, India without sharing the associated diseases in the diseasomics knowledge graph. All these three clinicians had 2 years of experience following their MBBS medical training and compulsory rotating internship in the hospital. The same list of 100 symptoms combinations were also given to three health workers (two certified dentists and one certified nurse) who are pursuing MBA in Hospital and Healthcare Management at Pune, India for validation in a blinded manner. In the process of blinded validation, all the six healthcare professionals were requested to "make a list of the possible differential diagnoses for 100 virtual patients who presented with these symptom combinations". These six volunteers were allowed to refer to Google search engine and published literature. The variability between the results generated by the volunteers was very high. This was mainly due to the subjectivity associated with the interpretation of symptoms. Due to a very high variability, we failed to statistically quantify the results with respect to the diseasomics knowledge graph. This type of wide variability has been well documented at the primary care level–a study at Mayo Clinic found that second opinion discovered 88% misdiagnoses at the primary care [50].

Since we were not successful in quantifying the results with young healthcare providers in our first approach, we then attempted a validation with experienced tertiary care doctors and authors Dr Arnab Ghosh, MD, DNB, MNAMS, PhD scholar, and Dr Prantar Chakrabarti, MD, DM, DNB in the second approach. We doubled the number of random draws from 50 to 100 symptom combinations and extracted the associated diseases from the knowledge graph.

We randomly generated 100 two co-occurring symptoms combinations and 100 three co-occurring symptoms combination. We used these 200 co-occurring symptoms combinations to fetch the associated diseases from our knowledge graph. For two symptoms, random 30 combinations (out of 100) did not produce any result. Similarly, for three symptoms 45 combinations (out of 100) did not produce any result. The 70 double-symptom combination fetched 751 diseases and 55 triple-symptom combinations fetched 139 diseases. This 890 disease-list was used for manual validation. We considered these 890 instances that fetched disease from knowledge graph as positive cases and 75 instances that did not fetch any associated disease from the knowledge graph as negative cases.

Out of 751 positive double-symptom diseases, the machine could correctly make differential diagnoses in 674 cases; and, 77 incorrect interpretations were observed. Out of 139 positive cases of three symptoms, 131 cases were correctly diagnosed, and 8 cases were incorrect. This gave us 805 True-positive (TP) and 85 False-positive (FP) cases. The machine's diagnoses were marked wrong considering some conditions to be having random associations for example muscle pain with fever. Therefore, though the machine identified the association correctly, we labeled it "incorrect" considering a casual co-existence of symptoms. We labeled 9 out of 85 such observations "incorrect" giving benefit of doubt to the manual system of evaluation. Out of the 75 negative cases 64 combinations should produce some diagnosis which are missing in

the knowledge graph. 11 combinations of negative cases do not have any known disease association in literature. This gave us 64 False-negative (FN) and 11 (TN) True-negative. Table 4 summarizes the differential diagnosis results. The results are available in S1 Table.

Based on the table (Table 4) above we computed the overall accuracy and the F1 score:

The overall accuracy is: [(TP+TN)/(TP+TN+FP+FN)] = 0.84559

The F1 Score computes to: [2*TP /(2*TP + FP + FN)] = 0.91529

Symptom checkers and differential diagnostic tools are typically evaluated on the basis of clinical vignettes or patient cases with confirmed and known diagnoses [51,52]. Therefore, in addition to the random combination of symptoms we performed a third validation cycle based on given diseases and their symptoms. We randomly selected 25 diseases including COVID19 from a set of typical diseases presented to general practitioners across different geographies and listed all known associated symptoms as UMLS codes (see S2 Table). These symptoms were verified by Dr. Puja Chowdhury. One by one, the symptom sets were fed into the knowledge graph-based diagnostic tool for validation. When a disease was associated with a maximum of 3 symptoms we used all symptoms for differential diagnosis. In case of more than 10 symptoms, we used a random selection of 10 symptoms to construct a symptom set that could be used for validation. The details are given in supplementary file S1 Table.

## Node2vec analysis

Using node2vec we can take disease nodes or symptom nodes and discover similar/closest nodes. Node2vec can predict likely edges that are absent in the original graph. This property of node2vec is commonly known as link prediction. We present here the link prediction property of node2vec in the context of our knowledge graph. We used symptom code 'C0019079: blood regurgitation (Hemoptysis) [Sign or Symptom]' for link prediction. This in other words means–discover all those diseases that may have the hemoptysis symptom but are not already present in the knowledge graph. This also may be considered a validation step for the completeness of the input data and validation of the symptom-disease network.

We found many diseases and symptoms close to each other in the embedded space with various probabilities. We examined the disease-symptom network for C0019079 (hemoptysis) and got a list of diseases that are semantically similar or close in the embedded space. We removed all those diseases where hemoptysis is already present as a symptom in the knowledge graph. We presented 3 diseases from our knowledge graph that do not include any symptom of hemoptysis but are close to hemoptysis in the embedding space. This logically means that those diseases should have links in the knowledge graph with hemoptysis symptom. We did a literature search to confirm that the links predicted by node2vec are indeed correct. These cases are:

1. DOID:6510 | Name: lung occult squamous cell carcinoma | (probability: 0.7140144109725952) [53]

2. DOID:7045 | Name: basaloid lung carcinoma | (probability: 0.7124731540679932) [54]

3. DOID:5547 | Name: pulmonary artery choriocarcinoma | (probability: 0.6947743892669678) [55]

## Discussion

50 years ago, Julian Tudor Hart proposed "The Inverse Care Law" [56]. Tudor Hart observed that disadvantaged populations need more health care than advantaged populations, but in reality, disadvantaged population receive less health care [56]. For example, Sub-Saharan

Africa, with slightly more than 10% of the world's population, accounts for 25% of the burden of disease; 3% of the world's health resources and around 1% of the world's trained health workers to deal with this disease burden. North America in contrast, with around 5% of the world's population, has about 3% of the burden of disease, 25% of the world's health resources, and 30% of the world's health workers [57]. In this paper we have presented the diseasomics knowledge graph to address this challenge and to promote the vision of universal health coverage.

The biomedical knowledge has grown tremendously through biological experiments and clinical studies. The resulting bodies of knowledge helped unleash unknown, hidden, and tacit knowledge which was converted into explicit knowledge in form of ontologies. All these ontologies are published and available in the open domain for download. Some of this knowledge is accessible through the Web browser as well. However, typically they are only available in silos and can only be interpreted by researchers. We have combined all these rich diverse knowledge bodies into one single machine interpretable knowledge graph to construct the digital twin of the disease knowledge. In the case of symptom to disease knowledge we went even further. We used a diverse set of knowledge sources from symptom ontology, disease ontology, human symptom disease network, acute emergency care, Wikipedia, PubMed, to custom annotations and constructed a rich symptom-disease network to help with differential diagnosis at the point-of-care. The accuracy of this knowledge graph is 84.56% which over a period of time will improve further. This will help a health worker in underserved or remote areas where doctors are not present. Assuming mobile technology and basic mobile based internet services to have better penetration, a health worker can make a differential diagnosis and offer acute care where health facilities are not apt. This will also reduce the medical error and turnaround time for diagnosis at the primary care. This will increase productivity of care providers and reduce the disease burden. With the help of AIoT (Artificial Intelligence of Things), it will also increase the footprint of care while simultaneously reducing the cost of care [4,34]. We have built a mobile App that demonstrates how the knowledge graph can support healthcare workers in capturing medical records efficiently and to establish a precise diagnosis and automated documentation at remote underserved regions of the world [34].

As our knowledge graph performs the accurate coding in an automated fashion, the use of this knowledge graph will reduce the documentation load. This in effect will help to reduce or prevent the physician burnout. Clearly, this automated coding system will enhance the healthcare system in LMIC (Low- and Middle-Income Countries), where resource limited EHR systems should be fully machine interpretable. This will help remove the Caucasian bias in medicine. Because our solution provides interoperability of the coding systems, the knowledge graph can be integrated with a hospital EHR through ICD10 CM or the SNOMED CT subgraph.

Knowledge graphs help discover hidden knowledge, whereas, graph embedding helps discover unknown knowledge. This knowledge graph thus constructed with both hidden knowledge and unknown knowledge is released in the public domain [58] for further enhancement (see S3 Text). The knowledge graph can be enhanced with any additional knowledge from EHR or other sources like SNOMED CT. This knowledge will be used in clinics for better diagnosis as well as for health workers, medical teaching and research.

Clinical diagnosis involves the discovery of the causation of the illness. The machine interpretable knowledge graphs presented in this paper are associations between various conditions and do not imply causation. The knowledge graphs and the associated tools presented here can be used as a guide. This will be suitable for triaging and decision support about when it is a good time to seek medical help or a referral.

These tools are also effective as a search engine for disorders that share similar symptoms, e.g., it is very easy to produce a list of all diseases that share fever as a common symptom. The knowledge graph can also be used as a tool in medical studies and nursing training to compile rare diseases associated with a set of symptoms that could potentially be overlooked if the healthcare provider does not have the special experience. The knowledge graph produces a list of diseases that is ordered by prevalence/disease burden from South Asia. This can lead to confusions if life threatening diseases like rabies appear on the list that are very rare in Central Europe but unfortunately quite common in India. Future versions of our knowledge graph will be customized to produce a list of potential diseases ordered by their prevalence in other regions including Europe and the US.

Clinical diagnosis is a very complex process. Tools like the knowledge graphs presented here can assist and support the differential diagnostic process but do not replace the careful root cause analysis, elimination process, and gathering of further information about lifestyle and health history.

## Supporting information

**S1 Text. This supporting information includes the details of the biomedical data used in this paper.**
(PDF)

**S2 Text. This supporting information contains screenshots of ancillary medical knowledge that show the advantage of machine interpretable actionable medical knowledge (*physician digital twins*) that can be used by expert and non-expert health workers alike.**
(PDF)

**S3 Text. Instructions for installation and detailed access information for data availability.**
(PDF)

**S1 Table. This is the Result of the random Differential Diagnosis test. For the purpose of provenance of the 84.56% accuracy, we have added 'Reference', 'Remarks', and 'Description' in the results along with the symptoms and their associated diseases.**
(XLSX)

**S2 Table. This is the list of 25 randomly selected diseases with all associated symptoms as UMLS codes.**
(XLSX)

## Acknowledgments

Sakthi Ganesh of Cybernetic Care used the Diseasomics knowledge graph APIs and developed a PWA applications to access the knowledge graph from iPhone and Android smartphones. Dr Puja Chowdhury, MBBS, MRCP Scholar and Dr L. V. Simhachalam Kutikuppala, MBBS, manually curated some of the knowledge used in Diseasomics. Dr. med. Tobias Kochsiek, specialist for internal medicine, inflammatory bowel diseases and nephrology, experimented with the diseasomics knowledge graph and gave valuable feedback regarding the diagnostic process.

## Author Contributions

**Conceptualization:** Asoke K. Talukder.

**Formal analysis:** Asoke K. Talukder.

**Methodology:** Asoke K. Talukder.

**Validation:** Arnab Ghosh, Rakesh Biswas, Prantar Chakrabarti.

**Writing – original draft:** Asoke K. Talukder, Lynn Schriml, Roland E. Haas.

**Writing – review & editing:** Asoke K. Talukder, Lynn Schriml, Arnab Ghosh, Rakesh Biswas, Prantar Chakrabarti, Roland E. Haas.

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
