## [Decision Letter · Decision Letter 0]

11 Nov 2021

PDIG-D-21-00048

Diseasomics: Actionable Machine Interpretable Disease Knowledge at the Point-of-Care

PLOS Digital Health

Dear Dr. Haas,

Thank you for submitting your manuscript to PLOS Digital Health. After careful consideration, we feel that it has merit but does not fully meet PLOS Digital Health’s publication criteria as it currently stands. Therefore, we invite you to submit a revised version of the manuscript that addresses the points raised during the review process.

While the reviewers found the paper interesting, they identified several areas that would benefit from further revision. In particular, the reviewers suggested: providing a more in-depth review of the state of the art models in this area; evaluating the model against common benchmarks where possible; providing more detail on the app (or perhaps removing this altogether); providing the code that underpins the paper in a public repository.

We look forward to receiving your revised manuscript.

Kind regards,

Tom J. Pollard, Ph.D.

Academic Editor

PLOS Digital Health

Journal Requirements:

1. We ask that a manuscript source file is provided at Revision. Please upload your manuscript file as a .doc or .docx.

2. Please provide separate figure files in .tif or .eps format only, and remove any figures embedded in your manuscript file. Please ensure that all files are under our size limit of 20MB.

For more information about figure files please see our guidelines: https://journals.plos.org/digitalhealth/s/figures

3. Please amend detailed Financial Disclosure statement. If you did not receive any funding for this study, please simply state: “The authors received no specific funding for this work.”

4. Please update the completed 'Competing Interests' statement. If you have no competing interests to declare, please state: “The authors have declared that no competing interests exist.”

5. Please provide a complete Data Availability Statement in the submission form, ensuring you include all necessary access information or a reason for why you are unable to make your data freely accessible. Note that it is not acceptable for the authors to be the sole named individuals responsible for ensuring data access.

PLOS defines a study's minimal data set as the underlying data used to reach the conclusions drawn in the manuscript and any additional data required to replicate the reported study findings in their entirety. Any potentially identifying patient information must be fully anonymized. 

If your research concerns only data provided within your submission, please write “All data are in the manuscript and/or supporting information files” as your Data Availability Statement.

Reviewers' comments:

Reviewer's Responses to Questions

**Comments to the Author**

1. Does this manuscript meet PLOS Digital Health’s publication criteria? Is the manuscript technically sound, and do the data support the conclusions? The manuscript must describe methodologically and ethically rigorous research with conclusions that are appropriately drawn based on the data presented.

Reviewer #1: Yes

Reviewer #2: Partly

Reviewer #3: Partly

2. Has the statistical analysis been performed appropriately and rigorously?

Reviewer #1: Yes

Reviewer #2: I don't know

Reviewer #3: No

3. Have the authors made all data underlying the findings in their manuscript fully available (please refer to the Data Availability Statement at the start of the manuscript PDF file)?

Reviewer #1: Yes

Reviewer #2: Yes

Reviewer #3: Yes

4. Is the manuscript presented in an intelligible fashion and written in standard English?

Reviewer #1: Yes

Reviewer #2: Yes

Reviewer #3: Yes

5. Review Comments to the Author

Reviewer #1: The paper entitled "Diseasomics: Actionable Machine Interpretable Disease Knowledge at

the Point-of-Care" is an excellent presentation of the possibilities available to us through careful and thoughtful analysis of ontologies. The most important part is the exercise that presents the relationships among terms to aid cause and effect in disease discovery.

The graphic representation of disease and pharmacology and the analysis of the relationships shown is outstanding. The attempt to understand and normalize the complex terms in the UMLS under 'findings', 'signs and symptoms' is one of the best I have seen so far to put in use this complex area. Findings, signs and symptoms are complex and difficult to understand in the real world, and tend to be 'messy' as described by other authors. most informatics studies tend to discard them to avoid confusion and decrease the amount of manual annotation and work. Incorporating them in the study brings in a new level that clearly shows their importance."

It will be interesting to see next steps where LOINC terms are brought in the picture.

Reviewer #2: The study aims to bring together various knowledge sources to help with diagnosis decisions through a knowledge graph, eventually targeting the healthcare system of low and middle income countries.

The paper is well written and explains the process of aggregating multiple knowledge sources for clinical decision support. I will leave the assessment of the technical details to the other reviewers and will point out a few concerns I have in relation to the overall study.

- I appreciate the detailed presentation of all the information sources and the process to combine them. But, I think these could be presented in a more concise way perhaps with a big table.

- Using Wikipedia as the main source of information is very problematic. The rationale behind this choice should be explained clearly.

- The background section covers some prior work but a critical assessment of the current state of art in this information aggregation and generating a differential diagnosis is missing. For example, "The ‘Symptom Ontology’ provides the ontology of many symptoms but lacks the cross-references to diseases [25]. Xia et al. attempted to address this gap [26]." What should we understand from this statement, did Xia et al succeed? what did they achieve? Similarly, "Rotmensch et al. constructed a disease-symptom network of 156 diseases and 491 symptoms from emergency department patient visits [28]. Here for the first time we present 11,188 conditions that include signs, symptoms, findings, diseases, syndromes, and neoplasm with 10,579 associated diseases." Does this mean the number of conditions and diseases are the higher than any prior systems? Please describe the unique contribution of this study more clearly.

- Although I'm not an expert in the technical side of NLP methods, I think any results should be presented with the current benchmarks. The paper does not seem to provide these. Related to the evaluation, the paper states that "We randomly generated 100 two cooccurring symptoms combination and 100 three cooccurring symptoms combination." what's the reasoning behind these numbers?

- Mobile app part of the paper is very underdeveloped and I think there is no need to include it without any evaluation.

Reviewer #3: The introduction was too broad and nonspecific, particularly for this targeted journal and audience. Much more time should have been spent discussing previous work around differential diagnosis support systems (e.g. DXplain).

The validation of the tool was particularly suspect. It would be much more useful to be validated against clinical opinion

and practice rather than itself. As a specific tool for untrained providers it might be useful, but certainly not for trained clinicians.

6. PLOS authors have the option to publish the peer review history of their article (what does this mean?). If published, this will include your full peer review and any attached files.

**Do you want your identity to be public for this peer review?** For information about this choice, including consent withdrawal, please see our Privacy Policy.

Reviewer #1: **Yes: **Laritza M. Rodriguez MD, PhD.

Reviewer #2: No

Reviewer #3: No

---

## [Editor Report · Decision Letter 1]

10 Feb 2022

PDIG-D-21-00048R1

Diseasomics: Actionable Machine Interpretable Disease Knowledge at the Point-of-Care

PLOS Digital Health

Dear Dr. Haas,

Thank you for submitting your manuscript to PLOS Digital Health. After careful consideration, we feel that it has merit but does not fully meet PLOS Digital Health's publication criteria as it currently stands. Therefore, we invite you to submit a revised version of the manuscript that addresses the points raised during the review process.

We look forward to receiving your revised manuscript.

Kind regards,

Tom J. Pollard, Ph.D.

Academic Editor

PLOS Digital Health

Additional Editor Comments (if provided):

Many thanks for submitting a revised version of the paper. While the changes go some way to address the reviewers' concerns, my feeling is that there are several aspects of the work that would benefit from improvement, in particular:

- The knowledge graph and associated software do not appear to be available in an open source repository. The Data Availability statement says that "The NeoImportDO.zip self-loading knowledge graph can be downloaded from https://cyberneticcare.com", but it is unclear where on this website the data can be downloaded from and I would not consider this a suitable long term repository. Instead I suggest the data and software should be deposited in a repository such as Zenodo or Figshare with a formal citation and persistent link (DOI).

- There is a demo of the prediction tool at: https://triage.cyberneticcare.com/diseasePrediction but there appears to be no documentation and I was unable to successfully generate a prediction using the tool. 

- Given that model validation is perhaps the most important aspect of the paper, I think significantly more effort is needed in this area. Currently the validation is described at a very high level, with limited details of the approach and results.

- In addition, the language in the validation section often lacks clarity (e.g. it is unclear to me exactly what is meant by "Monte Carlo methods are a class of computational algorithms that rely on repeated random sampling to compute quantities of interest without bias"...presumably "compute quantities" means estimating parameters? An unbiased estimate of a parameter requires an unbiased method to estimate the parameter -- using MC is not sufficient).

- Making the code and data used in the model analysis/validation available in a public repository would be very helpful. As far as possible, these resources should be organised in a way that demonstrates the approaches and allows them to be reproduced.
---

## [Decision Letter · Decision Letter 2]

14 Jul 2022

PDIG-D-21-00048R2

Diseasomics: Actionable Machine Interpretable Disease Knowledge at the Point-of-Care

PLOS Digital Health

Dear Dr. Haas,

Thank you for submitting your manuscript to PLOS Digital Health. After careful consideration, we feel that it has merit but does not fully meet PLOS Digital Health's publication criteria as it currently stands. Therefore, we invite you to submit a revised version of the manuscript that addresses the points raised during the review process.

Please submit your revised manuscript within 30 days Sep 12 2022 11:59PM. If you will need more time than this to complete your revisions, please reply to this message or contact the journal office at digitalhealth@plos.org. Please include the following items when submitting your revised manuscript:

We look forward to receiving your revised manuscript.

Kind regards,

Tom J. Pollard, Ph.D.

Academic Editor

PLOS Digital Health

Journal Requirements:

Additional Editor Comments (if provided):

Thanks for responding to the points raised by reviewers. On reading the updated version, I think it is important that the limitations and potential risks of the tool should be considered. I would suggest both (1) clarifying in the abstract that there are important limitations of the knowledge network (2) explicitly discuss limitations and risks in the discussion section, and how these might be mitigated. For example, if I search for "fever" at https://triage.cyberneticcare.com/diseasePrediction, the first result is "Rabies". What are the negative implications of offering (most likely) incorrect predictions?

Reviewers' comments:

Reviewer's Responses to Questions

**Comments to the Author**

1. If the authors have adequately addressed your comments raised in a previous round of review and you feel that this manuscript is now acceptable for publication, you may indicate that here to bypass the “Comments to the Author” section, enter your conflict of interest statement in the “Confidential to Editor” section, and submit your "Accept" recommendation.

Reviewer #1: All comments have been addressed

2. Does this manuscript meet PLOS Digital Health’s publication criteria? Is the manuscript technically sound, and do the data support the conclusions? The manuscript must describe methodologically and ethically rigorous research with conclusions that are appropriately drawn based on the data presented.

Reviewer #1: Yes

3. Has the statistical analysis been performed appropriately and rigorously?

Reviewer #1: Yes

4. Have the authors made all data underlying the findings in their manuscript fully available (please refer to the Data Availability Statement at the start of the manuscript PDF file)?

Reviewer #1: Yes

5. Is the manuscript presented in an intelligible fashion and written in standard English?

Reviewer #1: Yes

6. Review Comments to the Author

Reviewer #1: Clinical diagnosis is a complex process based not solely on symptoms but on signs and symptoms, paraclinical studies and physical exam. However the combination of resources and the model presented here is impressive. There have been earlier attempts based also on the UMLS and Bayes (Homero, and a couple other systems) , abandoned from clinical use because of it was not specific enough and clinicians do not really need general diagnosis tools like this. It would be best to promote it as a layman's tool to decide when is a good time to seek medical help. 

All comments were addressed properly.

7. PLOS authors have the option to publish the peer review history of their article (what does this mean?). If published, this will include your full peer review and any attached files.

**Do you want your identity to be public for this peer review?** For information about this choice, including consent withdrawal, please see our Privacy Policy.

Reviewer #1: Yes: Laritza M Rodriguez MD, PhD

---

## [Editor Report · Decision Letter 3]

14 Sep 2022

Diseasomics: Actionable Machine Interpretable Disease Knowledge at the Point-of-Care

PDIG-D-21-00048R3

Dear Dr. Haas,

We are pleased to inform you that your manuscript 'Diseasomics: Actionable Machine Interpretable Disease Knowledge at the Point-of-Care' has been provisionally accepted for publication in PLOS Digital Health.

Best regards,

Tom J. Pollard, Ph.D.

Academic Editor

PLOS Digital Health